# Comparative Analysis of Transcriptome and Proteome Revealed the Common Metabolic Pathways Induced by Prevalent ESBL Plasmids in *Escherichia coli*

**DOI:** 10.3390/ijms241814009

**Published:** 2023-09-12

**Authors:** Chuan Huang, Hoa-Quynh Pham, Lina Zhu, Rui Wang, Oi-Kwan Law, Shu-Ling Lin, Qi-Chang Nie, Liang Zhang, Xin Wang, Terrence Chi-Kong Lau

**Affiliations:** 1Department of Biomedical Sciences, College of Veterinary Medicine and Life Science, City University of Hong Kong, Hong Kong SAR, China; chuahuang4-c@my.cityu.edu.hk (C.H.); hqpham2-c@my.cityu.edu.hk (H.-Q.P.); linazhu3-c@my.cityu.edu.hk (L.Z.); rwang46@cityu.edu.hk (R.W.); carmenlaw918@gmail.com (O.-K.L.); sllin8-c@my.cityu.edu.hk (S.-L.L.); qcnie2-c@my.cityu.edu.hk (Q.-C.N.); liangzhang.28@cityu.edu.hk (L.Z.); 2Tung Biomedical Sciences Centre, City University of Hong Kong, Hong Kong SAR, China; 3Department of Surgery, Faculty of Medicine, The Chinese University of Hong Kong, Hong Kong SAR, China; xwang@surgery.cuhk.edu.hk

**Keywords:** ESBL plasmids, transcriptome, proteome, metabolic pathways

## Abstract

Antibiotic resistance has emerged as one of the most significant threats to global public health. Plasmids, which are highly efficient self-replicating genetic vehicles, play a critical role in the dissemination of drug-resistant genes. Previous studies have mainly focused on drug-resistant genes only, often neglecting the complete functional role of multidrug-resistant (MDR) plasmids in bacteria. In this study, we conducted a comprehensive investigation of the transcriptomes and proteomes of *Escherichia coli* J53 transconjugants harboring six major MDR plasmids of different incompatibility (Inc) groups, which were clinically isolated from patients. The RNA-seq analysis revealed that MDR plasmids influenced the gene expression in the bacterial host, in particular, the genes related to metabolic pathways. A proteomic analysis demonstrated the plasmid-induced regulation of several metabolic pathways including anaerobic respiration and the utilization of various carbon sources such as serine, threonine, sialic acid, and galactarate. These findings suggested that MDR plasmids confer a growth advantage to bacterial hosts in the gut, leading to the expansion of plasmid-carrying bacteria over competitors without plasmids. Moreover, this study provided insights into the versatility of prevalent MDR plasmids in moderating the cellular gene network of bacteria, which could potentially be utilized in therapeutics development for bacteria carrying MDR plasmids.

## 1. Introduction

Bacteria and plasmids have co-existed and co-evolved for a very long time, and plasmid–host interactions play an indispensable role in the evolution of microbes. Their interactions are much more complex than just conferring drug resistance through plasmid-borne resistance genes. While several studies have explored the role of MDR plasmids in bacterial resistance [1,2], few have delved into their comprehensive influence on gene expression and metabolic pathways [3,4]. Notably, environmental pressure is one of the major natural selection factors, and plasmids can mediate a multitude of adjustments in bacteria by facilitating plasmid-borne adaptive abilities and even rapid evolution [5,6]. Some plasmids encode genes that can assist bacterial hosts in utilizing alternative nutrient sources, like nitrogen, in case of a shortage or if the fixation of essential elements is needed [7]. Admittedly, plasmids also act like parasites to a certain extent since they entirely replicate for their own interests. Nevertheless, many mutualistic traits were also found in plasmids, from enhancing host virulence [8] to helping hosts produce necessary amino acids [9]. In return for these benefits, bacterial hosts generally provide all the required resources needed for plasmids to maintain and replicate. Nevertheless, the fitness cost is generally low and easily outweighed by favorable plasmid-borne traits. 

Beta-lactams are essential antibiotics that target bacterial cell walls. However, the emergence of extended-spectrum beta-lactamases (ESBLs) has reduced the clinical value of these powerful antibiotics [10,11]. Therefore, infection of *Enterobacteriaceae* with ESBLs often poses a significant challenge to public health globally. Plasmids with one or more drug-resistant gene(s) that confer multidrug resistance are called multidrug-resistant (MDR) plasmids. Dissemination of MDR plasmids is the most common pathway to spread antibiotic resistance among bacteria [12,13,14,15,16,17]. ESBL plasmids were identified as the most prevalent plasmids in the last few decades, and they were classified as incompatibility group (Inc) plasmids including IncF, IncI, IncL, IncM, IncN, and IncX [18]. Currently, over 1000 MDR plasmids from *Enterobacteriaceae* have been identified as these incompatibility groups, and they are commonly found in drug-resistant bacteria. 

The presence of plasmids in a bacterial host can significantly alter the genetic regulatory network [19]. During environmental changes, bacteria commonly adapt through the conjugative transfer of plasmid DNA and post-transcriptional regulation [20]. The physiology of drug-resistant bacteria has been extensively studied in the last few decades [21,22]. However, little is known about the influence of the prevalent MDR plasmids and their interaction with the bacterial genome, in particular, the survival advantages conferred to the bacteria in addition to providing drug resistance. Therefore, in this study, we investigated the functional roles of six major multidrug-resistant (MDR) plasmids including four ESBLs plasmids and two carbapenemases plasmids clinically isolated from patients. Moreover, the gene regulatory network mediated by these MDR plasmids and their interaction with the host genomes were determined at transcriptomic and proteomic levels. One of the highlights of this study was the discovery of the common MDR plasmids-induced pathways in the bacterial host, which were crucial for the survival of drug-resistant bacteria and could potentially be utilized as druggable targets for therapeutic development in the future. Moreover, the activation of various pathways by specific types of MDR plasmids was unveiled, which play significant roles in the adaptation of bacteria in different environments.

## 2. Results

### 2.1. Fitness and Persistence of Prevalent MDR Plasmids

To investigate the impact of multidrug-resistant (MDR) plasmids on bacterial fitness and their stability in the host, the growth of *Escherichia coli* strains J53 conjugated with different prevalent MDR plasmids was measured and compared (Table 1). As shown in Appendix A, we found that the doubling time in the exponential phase for all transconjugants ranged between 33 and 49 min, while that of isogenic J53 was 33.99 min, indicating the general fitness cost of MDR plasmids in *Escherichia coli* regarding cell growth. Since the presence of MDR plasmids caused fitness burdens, the persistence (stability) of these MDR genetic elements in the bacterial host was investigated using plasmid stability tests. As shown in Appendix A, all MDR plasmids were retained in the transconjugants after 600 generations without any selection pressures except the pNDM-HK plasmid (incompatibility group plasmid IncM2), which started to lose the plasmid in the 100th generation. Among all different Inc plasmids, intriguingly, the gene encoding the HN-S protein is absent in the pNDM-HK plasmid. Previous studies demonstrated that the HN-S protein is crucial for bacterial fitness, and deletion of HN-S protein genes on NDM1-carrying plasmids resulted in decreased plasmid stability [23,24]. Thus, it is plausible that the instability of the pNDM-HK plasmid is attributed to the missing HN-S protein.

### 2.2. RNomics Study to Evaluate the Impact of MDR Plasmids on the Host

Building on the findings related to bacterial fitness and plasmid stability, we next delved into the transcriptomic alterations induced by major MDR plasmids, including IncI, IncF, IncM2, and IncX. Using *Escherichia coli* J53 as a consistent genetic background, we introduced various plasmids individually via conjugation and conducted an RNA-seq analysis (Table 1 and Appendix A). Then, the transcriptomes of J53 and transconjugants were sequenced at the log phase using an Illumina NextSeq500 RNA-seq platform. For each sample, biological triplicates were performed. Overall, over 90% of the sequencing reads were mapped to the *Escherichia coli* reference genome (GenBank accession: NC 000913.3) for each sample (Appendix A), and the transcriptomes were compared with that of the isogenic J53 strain. The comparative transcriptomes are shown in Figure 1. Out of 4469 genes on the *Escherichia coli* J53 chromosome, approximately 2% of the genes displayed significant differences in expression (log2-fold change greater than one and TPM larger than 10) in the transconjugants (on average, 1% downregulated and 1% upregulated). Among the MDR plasmids, the presence of pHK01 and pNDM-HM380 plasmids had the highest effect on the gene expressions of the host (149 and 191 genes for J53/pHK01 and J53/pNDM-HM380, respectively, *p* < 0.01), while IncI group plasmids, pCTXM123_C0996 and pCTXM64_C0967, had the least impact (25 and 50 differentially expressed genes, respectively) on the host transcriptomes (Figure 2a). Moreover, DESeq2 was used to analyze the expression patterns in all transcriptomes, and we found fifty-four differentially expressed chromosomal genes in all transconjugants with a *p*-value < 0.01 (Figure 2b). The symmetrical distributions in the MA plots showing normalized reads versus log2-fold changes for all samples are shown in Appendix A. Among these genes, interestingly, *lsrF*, *lsrR*, and *lsrG* (involved in Autoinducer 2 degradation and biofilm formation), *IbsC* (the toxic peptide disrupting membrane integrity), and *rpoS* (stress-induced RNA polymerase sigma S factor) were all upregulated, suggesting the functional importance of these genes in the fitness of all MDR plasmids. In terms of individual plasmid-specific analysis (Appendix A), galactose salvage pathway genes *galP*, *galM*, and *galT* were downregulated in all transconjugants except J53/pCTXM123_C0996. This suggested that this particular transconjugant may utilize different metabolic pathways for the host nutrients compared with the transconjugants carrying other drug-resistant plasmids. Moreover, on the other hand, the anaerobic glycerol-3-phosphate dehydrogenase operon *glpABC* and *glpD* were upregulated only in J53/pCTXM123_C0996. Glycerol-3-phosphate is vital for bacterial virulence factor production, while *glpD* is essential for *Escherichia coli* respiration and metabolism when glycerol is the carbon source [25,26].

### 2.3. The Physiological Impacts of Differentially Expressed Genes Using Sample-Level Enrichment Analysis (SLEA)

To investigate the fitness of the plasmids and the physiological influence on the host, sample-level enrichment analysis (SLEA) with KEGG pathways was used to compare transcriptomes between *Escherichia coli* J53 and all the transconjugants (Appendix A). Low abundance genes (TPM < 5) were removed to avoid false positive gene expression. The overall differentially expressed patterns are shown in Appendix A. As shown in Figure 3a, the most upregulated pathway in all MDR plasmids was the tricarboxylic acid (TCA) cycle. The tricarboxylic acid cycle is one of the most important metabolisms in bacteria and is indispensable and required for bacterial fitness and virulence [27,28]. The upregulation of this pathway indicated a consensus effect imposed on the physiology of bacteria by the prevalent MDR plasmids for adaptation. Indeed, as shown in Figure 3b, the expression of the key genes in the TCA cycle, including *sdhA*, *sdhB*, *sdhC,* and *sdhD*, was increased significantly (about one-fold) in all samples except for transconjugants carrying the two IncX plasmids. Interestingly, the *sdhC* mutant has been shown to exhibit attenuated virulence in Gram-negative bacteria [29], and the upregulation of *sdhC* in bacteria carrying plasmids may reveal other functions of MDR plasmids than merely disseminating drug resistance genes. On the other hand, the most downregulated pathways were nitrogen metabolism, DNA replication, mismatch repair, homologous recombination, and biosynthesis, as shown in Figure 3c. All these pathways are energy-demanding, and the cells may be required to reduce their activity in order to accommodate the fitness cost of the plasmids. Noteworthy, the expression of nitrate reductase genes (*narG, narH, narK nib*, and *nirD*) was reduced by at least two-fold in J53/pHK01 and J53/pNDM-HK transconjugants (Figure 3d). As nitrate reductase genes are related to anaerobiosis, the observed down-regulation of anaerobic metabolism genes may imply the presence of an alternative metabolic mechanism for the utilization of resources to generate energy induced by these two plasmids. 

Among all the upregulated pathways, the control of the bacterial chemotaxis pathway in transconjugants carrying IncI plasmids (pCTXM123_C0996 and pCTXM64_C0967) was completely different from that of other groups of MDR plasmids (Figure 3e). Chemotaxis is essential for bacteria to sense and move to acquire food in different environments. The differences in regulating the chemotaxis pathway suggested that different plasmids may utilize different strategies to control bacterial mobility for nutritional sensing.

### 2.4. Mass Spectrometry Revealed the Variation in Proteomes from Bacteria Carrying Different MDR Plasmids

We performed a comparative proteomic analysis of *Escherichia coli* strains J53 and six transconjugants under the same conditions that we used in RNA-seq. Each sample was performed in biological triplicates and showed excellent reproducibility for each repeat (Appendix A). A total of 1330 chromosomal proteins were identified, covering 26.4% of the proteome of *Escherichia coli* K-12 (GENOME entry T00007), and 83.2% of the identified proteins (1107 proteins) were overlapped with the genes in transcriptomes (Figure 4a). The most abundant proteins were plasma proteins (740). The others included plasma membrane and outer membrane proteins (256 and 93, respectively) as well as 219 undetermined proteins. Moreover, a small number of nucleoid proteins were found (Figure 4b). In addition to the chromosomal proteins, we also identified plasmid-encoded proteins, which comprised less than 1% of the whole proteome and less than 10% of all predicted open reading frames (ORFs) on the plasmids. A list of the identified plasmid-encoded proteins is presented in Appendix A. 

Based on the functions of the plasmid-encoded proteins, we categorized them into different groups including plasmid maintenance, transposases, transporters, antibiotic-resistant enzymes, and uncharacterized proteins (Appendix A). The plasmid maintenance proteins consisted of the replicator (*repA*), plasmid partition system (*parA, parB*), components of the conjugation machinery (*traL, traU, traC*, and *virB*), and an addiction module (*pemK*). Notably, a high abundance of all antimicrobial-resistant enzymes, such as beta-lactamases, were identified in all transconjugants, even when the bacteria were not cultured under any stress conditions. The transporter systems found in the proteomes of pCTXM64_C0967 and pHK01 were nickel permeases (NikB, NikC) and an iron transporter (EitA), respectively. Interestingly, some uncharacterized proteins that possess functional domains such as a DNA-binding domain (HTH-type transcriptional regulator AB185_RS01320 in pNDM-HN380) or an RNA-binding domain (YafB in pCTXM123_C0996 and YaeC in pCTXM64_C0967) were found. As these proteins were found in abundance, they may provide significant function to the host under the control of these plasmids specifically.

To gain an overview of the six proteomes, volcano plots with fold changes and *p*-values were plotted for all identified chromosomal proteins. As shown in Figure 5a, only 2–5% of the proteins were identified as differentially expressed proteins (DEPs) which showed significant up- or down-regulation with an absolute fold change larger than 1.5 (*p*-value of ≤ 0.05) compared with the wild-type J53 (Appendix A). A small number of DEPs indicated a minor impact of MDR plasmids on the proteome of the bacterial host under unselected conditions. To further investigate the impact of MDR plasmids on the host cells, we performed a hierarchical clustering analysis of all DEPs among six transconjugants compared with the original host strain, J53, and presented the results as a heatmap. Among the 55 DEPs, the expression of 38 proteins was upregulated in all six strains, of which 12 were the same among six transcriptomes (Appendix A). Interestingly, although the number of DEPs varied among the strains, the heatmap showing the upregulated genes revealed a common pattern of expression (Figure 5b). The most significant upregulated genes across all six transconjugants include *narJ, tdcD, tdcB, tdcG, garD, garL, glpB, nanT, dsmA, dsmB*, and *ychN*. NarJ is the nitrate reductase 1 molybdenum cofactor assembly chaperone, a component of nitrate anaerobic respiration. The three genes *tdcD, tdcB*, and *tdcG* belong to the anaerobic threonine/serine degradation pathway in the absence of glucose, while GarD/GarL are involved in catalyzing the dehydration of galactarate. DmsA and DmsB are subunits of dimethyl sulfoxide (DMSO) reductase, which allows *Escherichia coli* to grow anaerobically on DMSO as a respiratory oxidant. Interestingly, in contrast with the transcriptome data, GlpB, which is encoded by the *glpABC* operon, was found to be upregulated in all six plasmids. 

For the downregulated DEPs, nine proteins were identified in most of the proteomes (Appendix A). FbaB (fructose-bisphosphate aldolase class I) is related to glycolysis and gluconeogenesis, while AceA/AceB (isocitrate lyase/malate lyase) are key enzymes in the glyoxylate cycle, a bypass for the TCA cycle when two-carbon compounds such as acetate are used as the sole carbon source. We also observed the downregulation of proteins specifically involved in branched-chain amino acid transport and nitrogen assimilation, such as GlnE and LivJ, as well as the aspartate-chemotaxis protein Tar. Notably, the ssDNA-specific exonuclease RecJ was decreased across all six transconjugants. Conjugation is indeed the mechanism that leads to the transient occurrence of ssDNA in the recipient cell; therefore, repression of the host RecJ might inhibit the entry of other plasmids DNA.

### 2.5. The Presence of ESBL Conjugative Plasmids Increased the Metabolic Pathway Related to Carbon Utilization at the Translational Level

To highlight the common metabolic pathways influenced by the six MDR plasmids, we analyzed the 55 DEPs with the STRING database, URL: https://string-db.org/ (accessed on 4 December 2022), to predict and visualize the protein–protein interaction (PPI) network (Figure 6). We found that numerous proteins clustered to the PPI networks, which involved in the utilization of different carbon sources such as anaerobic respiration, L-threonine catabolism to propionate (*tdc* operon), galactarate catabolism (*gar* operon), sorbitol PTS permease (*srl* operon), and N-acetylglucosamine catabolism (*nan* operon). Moreover, except for *aceA* and *aceB*, all the clustered proteins were upregulated compared with J53, indicating that those functional pathways were important and specific to the ESBL plasmids. For instance, the anaerobic respiration in the PPI consists of anaerobic redox enzymes that produce precursor molecules for the TCA cycle (*glpB, glpA, glpX*) or transfer electrons to terminal acceptors such as nitrate (*narG, narx*) or DMSO (*dmsA, dmsB*). *garD* and *garPLRK* are two transcriptional units of D-galactarate (*gar*) metabolism in *Escherichia coli*. The *tdc* and *nan* operons are required for utilizing L-threonine/serine and sialic acid from the environment, respectively. Noteworthy, both the *tdc* and *nan* operons are induced under anaerobic conditions. The promoter of the *tdc* operon is regulated by FNR, the primary transcriptional regulator for anaerobic metabolism [30], whereas sialic acid is highly present in the secreted mucus covering the intestinal lumen [31]. These functional pathways indeed attribute survival advantages to bacteria during infection.

## 3. Discussion

Previous studies of emerging drug resistance were mainly focused on the dissemination of resistance genes among the bacteria. However, the influence of prevalent MDR plasmids on gene expression in bacteria remains unclear, in particular, the contribution of plasmid-encoded non-resistance genes to the survival advantages of bacteria during infection. In this study, we determined and compared the global gene expression of *Escherichia coli* influenced by six prevalent MDR plasmids, pCTXM123_C0996 (IncI1), pCTXM64_C0967 (IncI2), pHK01 (IncFII), pNDM-HK (IncM2), pNDM-HN380 (IncX3), and pJIE143 (IncX4), at both transcriptomic and proteomic levels. RNA-seq analysis revealed that MDR plasmids led to significant changes in the expression of specific genes related to metabolic pathways in the bacterial host, whereas mass spectroscopic analysis illustrated a more limited impact on the proteomes of each transconjugant. This observation is attributed to the dynamic control of transcriptomes with the subtle regulation of proteomes to accommodate physiological changes in the bacterial host. A key finding of the analysis was the consensus effect imposed by different drug-resistant plasmids on bacteria, including the upregulation of genes in the tricarboxylic acid (TCA) cycle, which is the crucial pathway for bacterial fitness and virulence. The upregulation of genes in the tricarboxylic acid (TCA) cycle, a crucial pathway for bacterial fitness and virulence, suggests that MDR plasmids might confer a metabolic advantage to host bacteria. This enhanced energy production could potentially aid in the rapid response to antibiotic stress, thereby contributing to increased resistance. Furthermore, the modulation of oxidative phosphorylation, which is pivotal for ATP generation, suggests a potential role for energy-dependent drug pumps in antibiotic resistance [32]. However, it is important to note that direct evidence linking plasmid presence to the expression or activity of these pumps is currently lacking in this study, and further research is needed to substantiate this hypothesis. Moreover, the transcriptomic data revealed that different plasmids may use specific strategies to regulate nutrient metabolism, providing alternative pathways to enhance the adaptation of bacteria. This finding advances our understanding of MDR plasmid effects, showcasing their broader influence beyond what has been previously reported in the literature. Our study underscores the importance of considering the multifaceted roles of MDR plasmids, not just their contribution to drug resistance.

In addition to TCA cycles, the oxidative phosphorylation pathway was also upregulated in all transconjugants. Oxidative phosphorylation generates ATP, which provides energy for bacterial life processes including the expression of resistance genes and the replication and transmission of plasmids. Some MDR plasmids carry genes encoding energy-dependent drug pumps, which can expel antibiotics from the cell, thereby conferring antibiotic resistance to the bacteria. This process requires ATP as an energy source, which is dependent on oxidative phosphorylation. Some studies found that molecules involved in the process of oxidative phosphorylation such as ATP synthase may play a role in regulating bacterial resistance. For instance, the inhibition of ATP synthase could potentially decrease bacterial resistance [33,34]. Moreover, antibiotics such as streptomycin inhibit oxidative phosphorylation to kill bacteria.

In our study, we used a comparison between transcriptomes and proteomes to shed light on the cellular control of gene expression. Our findings revealed discrepancies between these two levels, underscoring the existing gaps and complexities in understanding the relationship between gene expression and protein production. For example, the *glpABC* operon, which is involved in anaerobic glycerol-3-phosphate metabolism, was found to be upregulated in all six transconjugants at the proteomic level, whereas only enhanced transcripts were observed in J53/pCTXM123_C0996 transconjugants, suggesting additional regulatory mechanisms such as post-transcriptional or pre-translational regulation. Indeed, regulatory small RNAs that target either the stability of mRNAs or the translation process have been shown to play crucial roles in this regard [35].

Another interesting finding of this study was that MDR plasmids can reprogram the carbon metabolism and anaerobic respiration of bacterial host cells including *tdc*, *nan*, *gar*, and *srl* operons. These operons are involved in the utilization of different carbon sources (L-serine, L-threonine, galactarate, sialic acid, and sorbitol) in anaerobic environments, which are crucial for bacterial adaptation in the gut environment. Several lines of evidence have shown that disruption of the *gar* and *nan* operons impair the colonization of *Escherichia coli* and *S. enterica* serovar Typhimurium in the mouse intestine [36]. Moreover, the metabolism of galactarate produced by host-mediated oxidation of galactose and the utilization of L-serine have been shown to be advantageous for bacterial growth and adaptation in the gut [37,38]. In addition, the utilization of sorbitol as a carbon source (*srl* operon) has been described in Firmicutes, which are the main components of gut microbiota [39,40]. Based on these findings, we speculated that MDR plasmids can contribute the growth advantages to bacterial hosts in the gut, leading to the expansion of plasmid-carrying bacteria over competitors without plasmids. The gut microbiota is the reservoir of *Enterobacteriaceae* that provides a favorable site for horizontal dissemination of conjugative plasmids. Plasmids that persist in the gut may have a greater chance of moving among different strains and exchanging genetic materials, resulting in the acquisition of antibiotic-resistance genes. Our findings on the differential expression of genes related to carbon utilization, such as the *tdc*, *nan*, *gar*, and *srl* operons, indicate a potential shift in metabolic preferences influenced by MDR plasmids. Such shifts could be adaptive strategies used by bacteria to thrive in environments with varied nutrient availability, further contributing to their fitness and survival in the presence of antibiotics. Noteworthy, we identified several MDR plasmid-specific genes that could have therapeutic implications (Appendix A). For instance, genes like *dmsA* and *dmsB*, which are subunits of dimethyl sulfoxide (DMSO) reductase, could be targeted to disrupt anaerobic respiration in *Escherichia coli.* Similarly, EitA, an iron transporter, could be a target to interfere with essential metal ion transport. These are preliminary insights and would require further experimental validation for their druggability. From a clinical perspective, understanding the metabolic advantages conferred by MDR plasmids can inform therapeutic strategies. If bacteria carrying MDR plasmids have enhanced energy production, this could be a potential target for novel treatments. Drugs that disrupt these metabolic pathways might make bacteria more susceptible to existing antibiotics, offering a combination approach to tackle antibiotic resistance.

In the transcriptomic data, the expression level of *rpoS* increased in all transconjugants at the log phase (RpoS was undetectable in the proteomic data). Since RpoS is a master regulator that coordinates the general stress response in starvation, oxidative stress, acidic stress, heat stress, and osmotic stress in *Escherichia coli*, a fitness burden was probably introduced by the presence of MDR plasmids, even in a stress-free environment. Indeed, we measured and compared the cell growth of J53 in the presence and absence of MDR plasmids to assess the fitness of transconjugants. As illustrated in Appendix A, the doubling time in the exponential phase for all transconjugants was slower than that of isogenic strain J53, indicating the fitness cost associated with MDR plasmids in *Escherichia coli*. Notably, plasmid-encoded virulence genes such as *spvABCD* in *Salmonella* spp. are regulated by RpoS [41]. Moreover, a recent study proposed that RpoS plays a vital role in the transformation of small non-conjugative plasmids. Overexpression of *rpoS* enhanced the transformation frequency [42]. Therefore, the upregulation of *rpoS* in our transconjugants could potentially contribute to the dissemination of such small plasmids, while it is worth noting that large conjugative plasmids may not be transmitted through this mechanism. Considering these factors, it is plausible to speculate that MDR plasmids enhance bacterial virulence and resilience against stress environments, albeit at the expense of growth rate. The observed transcriptional response to MDR plasmid acquisition, especially the upregulation of anaerobic metabolism, suggests a broader adaptive strategy beyond just antibiotic resistance. This adaptation might provide bacteria with the flexibility to survive in varied environmental conditions, further emphasizing the multifaceted roles of MDR plasmids in bacterial physiology.

The impact of MDR plasmids on bacteria host cells has been studied in terms of growth rate, motility, biofilm formation, stress response, and virulence. Some plasmids reduce fitness while others have no significant burden to the host. This variance may be attributed to different genetic backgrounds of bacterial hosts as well as the diversity of the plasmids. Interestingly, a study analyzing the effect of MDR plasmid pLL35 on various *Escherichia coli* strains showed a minor transcriptional response to its acquisition and upregulation of anaerobic metabolism across transconjugants [43]. The clinical ramifications of these findings extend beyond understanding bacterial physiology. If MDR plasmids can influence bacterial virulence and resilience, this could impact how infections are treated. For instance, treatment strategies might need to be adjusted to account for plasmid-carrying bacteria that are more virulent but grow slower. However, it is crucial to note that these are preliminary observations, and translating these into practical treatment strategies would require extensive further research. Therefore, while our study highlights the potential for MDR plasmids to affect treatment approaches, these implications should be considered as avenues for future research rather than immediate clinical applications. Recognizing the broader adaptive strategies used by bacteria carrying MDR plasmids can inform the development of more effective treatment regimens.

## 4. Materials and Methods

### 4.1. Strains and Plasmids Used in This Study

*Escherichia coli* (*Escherichia coli*) J53 was used as the host for conjugation of MDR plasmids [44]. The MDR plasmids used in this study are listed in Table 1. The clinical strains carrying MDR plasmids were kindly provided by Dr. P.L. Ho (The University of Hong Kong). The plasmids were transferred from the parental strains to *Escherichia coli* J53 by conjugation using the filter mating method. Briefly, the parental strains and J53 were cultured in LB overnight at 37 °C. Then, the medium was removed, and the bacteria were resuspended in a fresh LB medium. The donor and recipient were mixed in a 1:1 ratio and spotted onto a 45 µm filter membrane on an LB agar plate. After 1 h at 37 °C, the membrane with the bacteria was transferred to a new 15 mL Falcon tube containing 1 mL of fresh LB. The bacteria were resuspended from the membrane by vortexing. Transconjugants were selected by serial dilution and spotting on LB agar plates containing 100 µg/mL sodium azide and 10 µg/mL cefotaxime.

### 4.2. Growth and Doubling Time Measurement

Bacteria were cultured in LB broth at 37 °C with shaking (250 rpm). Growth was monitored every hour using the optical density until the readings reached a plateau. A growth curve of optical density versus time was plotted, and the doubling time was calculated. This experiment was carried out on biological triplicates. *Escherichia coli* J53 was used as a control.

### 4.3. Plasmid Stability Assay

In order to assess the stability and fitness of the conjugated bacteria, an in vitro assessment of plasmid stability and fitness cost following previously established methods was performed [23,45]. Briefly, 3 µL of an overnight growth of the bacteria in Luria–Bertani (LB) broth were inoculated into 3 mL of fresh LB broth and incubated for 12 h at 37 °C (time zero). The above process was repeated every 12 h (equivalent to 10 generations each). At time zero and after passage in the absence of antibiotics for every 20 generations (up to 600 generations), a sample of the culture was diluted. A series of dilutions was prepared and for each dilution, 100 µL were immediately plated onto a pair of plain and antibiotic-containing (50 ng/mL Cefotaxime) LB plates. Plasmid stabilities were determined using the percentages of colonial growth on the antibiotic-containing plates. This experiment was conducted in biological triplicates.

### 4.4. RNA Isolation

In the RNA-seq experiment, the transconjugants were cultured in standard Lysogeny broth (LB) medium and harvested at mid-log phase (OD_600_ = 0.6) in biological triplicates. *Escherichia coli* J53 was used as a control in all experiments. Total RNAs were extracted from the harvested cell pellets using a standard TRIZOL method (Thermo Fisher Scientific, Waltham, MA, USA). The TURBO DNA-free™ Kit (Thermo Fisher Scientific, Waltham, MA, USA) was used to remove host genomic DNA (gDNA). sRNA and mRNA were isolated from the total RNA using a MICROExpress™ Bacterial mRNA Enrichment Kit (Thermo Fisher Scientific, Waltham, MA, USA) followed by a Ribo-Zero rRNA Removal Kit (Illumina, San Diego, CA, USA). Both kits were used to remove ribosomal RNA. Sequencing libraries were constructed using a NEBNext^®^ Ultra™ Directional RNA Library Prep Kit for Illumina (Illumina, New England Biolabs, Ipswich, MA, USA) from the enriched sRNA and mRNA and RNA-sequencing was performed using an Illumina NextSeq system (NextSeq 500/550 Mid Output Kit v2, Illumina, San Diego, CA, USA).

### 4.5. Sequencing and Bioinformatic Analysis

SRA files generated from the Illumina NextSeq500 were converted to fastq files using the SRA Toolkit (accessed on 1 June 2019). The Bowtie2 alignment tool (version 2.3.1) was used to perform sequence alignments based on the converted fastq files and the Bowtie indexes of the *Escherichia coli* MG1655 genome (GenBank accession: NC 000913.3). SAMtools (version 0.1.19) was used to sort and compress aligned sequence data into BAM files. The sorted and compressed BAM files were then used for downstream processes in the Integrative Genomics Viewer (IGV) and R software (version 3.4.0) [46]. Two R packages, GenomicAlignments and GenomicFeatures were used to analyze and quantify gene expressions. Gitools, URL: http://www.gitools.org/ (accessed on 1 September 2019), was used to perform sample-level enrichment analysis (SLEA) of KEGG pathways. The RCircos package (accessed on 1 October 2019) was used to present an overview of the transcriptome data [47].

### 4.6. Filter-Aided Sample Preparation

Three single colonies of each strain were chosen to prepare the starters. Then, the bacteria were inoculated into fresh LB medium in a ratio of 1:100 and cultured at 37 °C with shaking (250 rpm). At OD_600_ = 0.6, the bacteria were harvested, washed off the medium, and resuspended in lysis buffer containing 8 M Urea, 100 mM Tris-HCl (pH = 8), 50 mM EDTA, and protease inhibitor cocktail (Roche). The bacteria were lysed with sonication at amplitude 20%, 5 s on/10 s off for 1 min. Then, cell debris were removed with centrifugation at the max speed for 15 min at 4 °C. The protein concentration was measured using Bradford. The cell lysate was loaded with a 3k-Da MWCO spin column (Millipore) and spun at 14 k × g for 30 min at room temperature. Then, 200 µL 8 M urea solution was added to the column for further washing by re-spinning the column. The proteins were alkylated by adding 100 µL 0.05 M iodoacetamide 8 M urea buffer. The columns were shaken at 800 rpm for 1 min and incubated in the dark at room temperature (25 °C) for 20 min before centrifugation at 14,000× *g* for 30 min. The proteins were washed again with the 8 M urea buffer twice. Subsequently, the buffer exchange was carried out using three washes with 100 µL 50 mM ammonium bicarbonate buffer. 

### 4.7. On-Filter Trypsin Digestion

Trypsin digestion was performed using 0.2 µg/μL sequencing grade modified trypsin (V5111, Promega, Madison, WI, USA) added to the column in the ratio 1:50 enzyme-to-protein. The reactions were incubated at 37 °C for 16 h with shaking at 600 rpm. The column was put on an Eppendorf tube and sealed with parafilm to prevent the column from drying out. In the next day, the columns were spun at 14,000× g for 30 min, and the digested peptides were collected in the collection tubes. Then, 5 µg peptide was desalted using C18 ziptip following the manufacture (Millipore, Burlington, MA, USA). Finally, the peptide pellet was dissolved in 12 µL of 0.1% formic acid and injected into the MS system.

### 4.8. High-Performance Liquid Chromatography Coupled with Tandem Mass Spectrometry (HPLC-MS)

All peptide samples were analyzed using the Q Exactive HF-X hybrid quadrupole-Orbitrap mass spectrometer coupled to an EASY-nLCTM 1200 system (Thermo Fisher Scientific, Waltham, MA, USA). For each sample, 2 µL of peptide mixture was resolved using an analytical C18 column (250 mm, 75 µm, 3 µm; PepSep, Denmark) at a flow rate of 250 nL/min for 75 min. The mobile phase was a mixture of buffer A and buffer B (0.1% formic acid in 80% acetonitrile) with a changing gradient of buffer B as follows: 0–2 min in 3–7%; 2–52 min in 7–25%; 52–62 min in 25–44%; 62–70 min in 44–95%; and 70–75 min in 95%. Spectrum recording was operated in the range of 350 to 1800 *m*/*z* with a mass resolution of 120,000. The positive ion mode was used with the spray voltage at 2500 V and a spray temperature of 320 °C. The resolution of dd-MS2 was 30,000 with a 1e5 of AGC target. The maximum IT was set at 60 ms, and the loop count was 12. The isolation window was 1.6 *m*/*z*, and the fixed first mass was 120.0 *m*/*z*.

### 4.9. Data Processing

The Uniprot *Escherichia coli* MG1655 (20 April 2019 download) database was used. The MS/MS data were processed using Maxquant 1.5.2.8 software, and the identification parameters were set with a precursor ion mass tolerance of ±10 ppm, fragment ion mass tolerance of ±0.02 Da, maximum of two missed cleavages, static modification with carboxyamidomethylation (57.021 Da) of the Cys residues, and dynamic modification with oxidation modification (+15.995 Da) of the Met residues. The peptide length ranged from 6 to a maximum of 144. According to the primary data analyses, the protein and peptide false discovery rates (FDRs) were set to 0.05 (5%), the decoy database was set to revert, and the minimum peptide number required for protein identification was one. Data with *p* ≤ 0.05 and a difference ratio of ≥1.2 were selected for further analysis. The statistical analyses of the LC-MS/MS data were performed using Perseus (v1.4.1.3).

### 4.10. Protein Quantitation and Analysis

Differential expression analysis was conducted between six transconjugants of J53 harboring different plasmids and the wild-type *Escherichia coli* J53 using the R package “limma”, respectively. 

### 4.11. Statistics

A paired Student’s *t*-test was performed to calculate the *p*-values. *p*-values < 0.05 were considered significant.

## Figures and Tables

**Figure 1 ijms-24-14009-f001:**
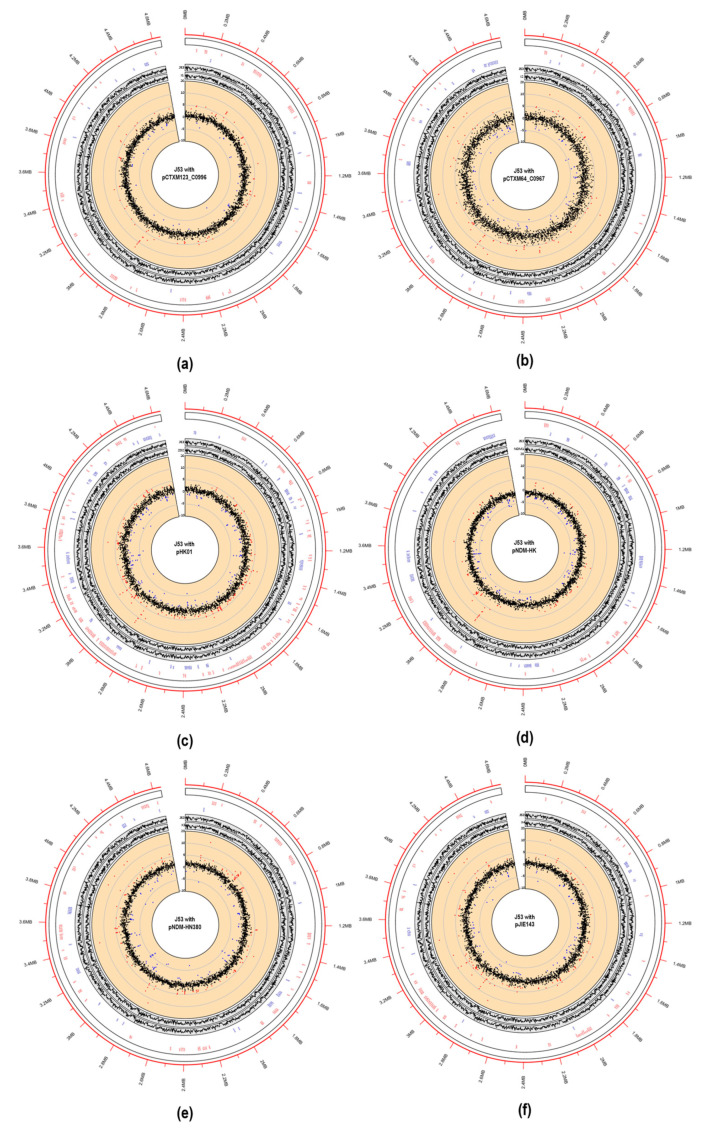
Transcriptome circles of *Escherichia coli* carrying MDR plasmids J53/pCTXM123_C0996 (**a**), J53/pCTXM64_C0967 (**b**), J53/pHK01 (**c**), J53/pNDM-HK (**d**), J53/pNDM-HN380 (**e**), and J53/pJIE143 (**f**). The outermost circle (in red) shows the genome coordinates (in Mbp) of J53. The first inner circle denotes the gene names of transconjugants, highlighting those with a log2-fold change greater than one (upregulated in red and downregulated in blue) compared with J53, as well as the genomic locations of these genes. The second inner circle demonstrates the gene expression of J53 and MDR plasmid transconjugants in TPM (transcripts per million) on the log2 scale. The third inner circle depicts the log2-fold change in gene expression of MDR plasmid transconjugants compared with J53. In this circle, the red and blue dots represent upregulated and downregulated genes (log2-fold change greater than one), respectively, while the other genes are colored in black.

**Figure 2 ijms-24-14009-f002:**
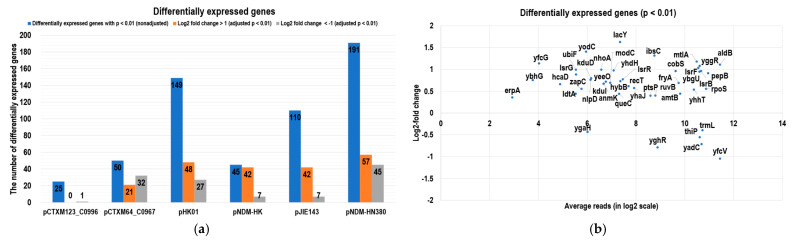
Differential gene expression of *Escherichia coli* carrying MDR plasmids. (**a**) Differentially expressed genes of each MDR plasmid transconjugant with adjusted and non-adjusted *p*-value < 0.01. Columns in blue represent the number of differentially expressed genes with a *p*-value less than 0.01 before applying the Bonferroni correction. Columns in orange represent the number of upregulated genes with an absolute fold change greater than one and an adjusted *p*-value less than 0.01. Columns in grey represent the number of downregulated genes with an absolute fold-change greater than one and an adjusted *p*-value less than 0.01. (**b**) A plot showing differentially expressed genes in all MDR plasmid transconjugants. The *X*-axis represents the average reads of all transconjugants on the log2 scale. The *Y*-axis represents the log2-fold change of the gene expression compared with J53.

**Figure 3 ijms-24-14009-f003:**
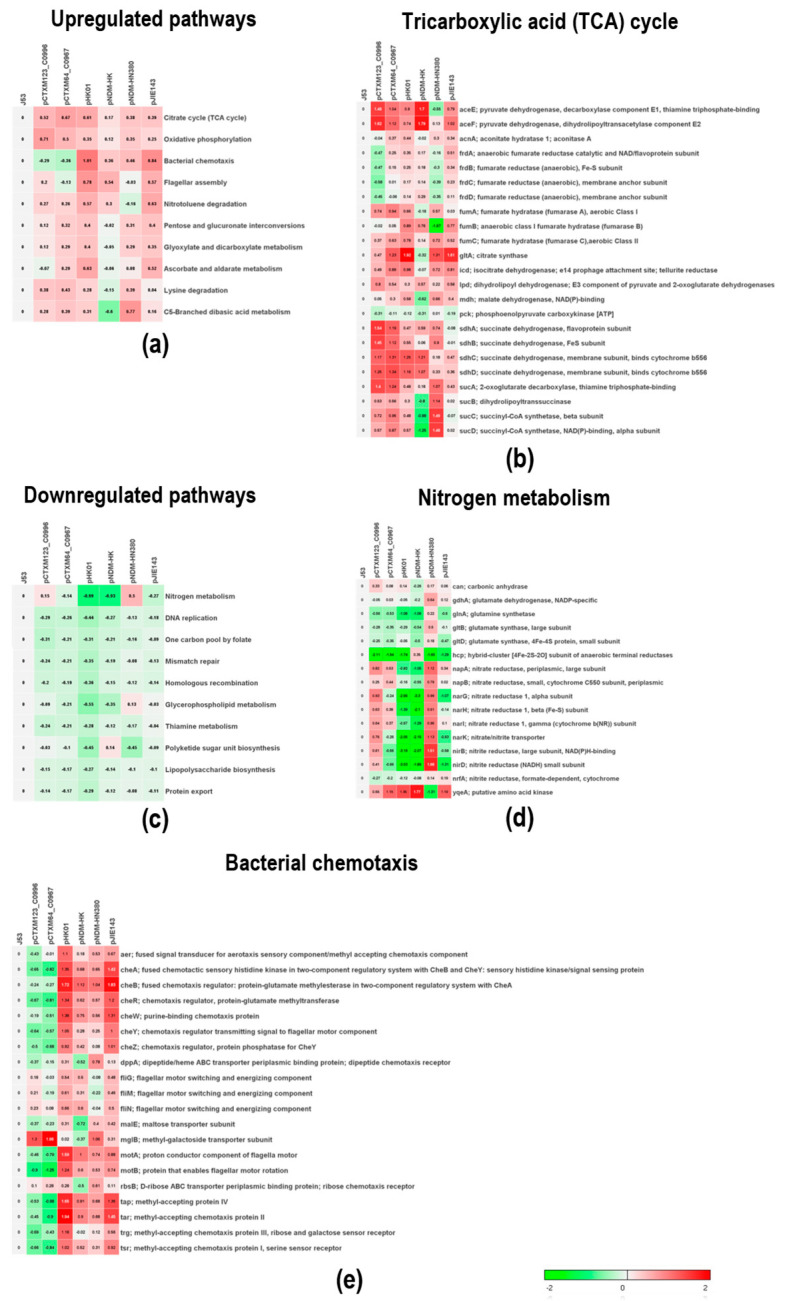
Heatmap showing the results of the sample-level enrichment analysis (SLEA). (**a**–**e**) Heatmaps for upregulated and downregulated pathways and genes in the bacterial citrate cycle (TCA cycle) pathway/the nitrogen metabolism pathway and the chemotaxis pathway, respectively. The number in squares represents the log2-fold change in the gene of the MDR plasmid transconjugant compared with J53. Upregulated genes are colored in red and downregulated genes are colored in green. The color scale is shown at the bottom right.

**Figure 4 ijms-24-14009-f004:**
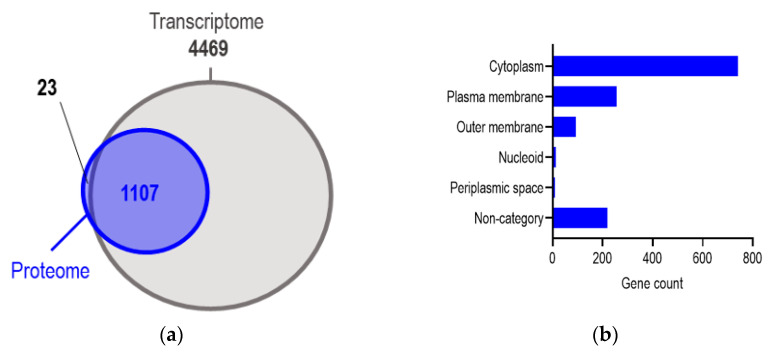
(**a**) A Venn diagram comparing the total number of proteins in the proteomic analysis with the total number of transcripts in RNA-seq among all six transconjugants. (**b**) Subcellular localization of proteins in proteomes.

**Figure 5 ijms-24-14009-f005:**
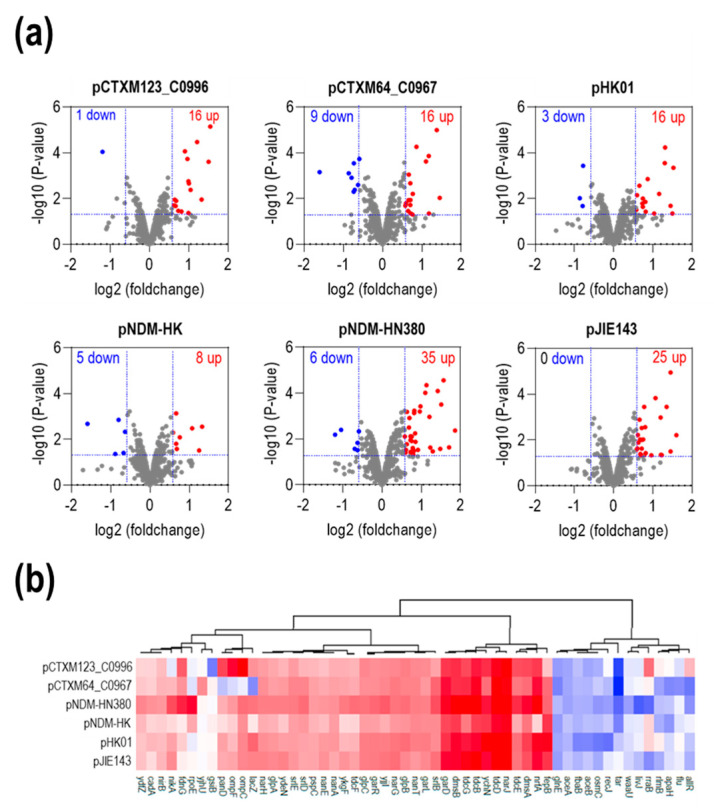
Mass spectrometry revealed the distinct protein profiles of six transconjugants harboring different MDR plasmids. (**a**) Volcano plots for six J53 transconjugants harboring different MDR plasmids (pCTXM123_C0996, pCTXM64_C0967, pHK01, pNDM-HK, pNDM-HN380, and pJIE143) compared with the wild-type J53 in the non-stress condition. Differentially expressed proteins (DEPs) are defined as those with *p*-values ≤ 0.05 and absolute fold change ≥ 1.5, corresponding to the rectangular regions. The left rectangular regions are the downregulated proteins, and the right rectangular regions are the upregulated proteins. The dots represent each identified protein. The upregulated DEPs are colored red, while the downregulated DEPs are colored blue. The other proteins are colored gray. (**b**) Heatmap of DEPs across all six transconjugants compared to the wild-type J53. The hierarchical clustering was performed using Euclidean distance and a Ward linkage model.

**Figure 6 ijms-24-14009-f006:**
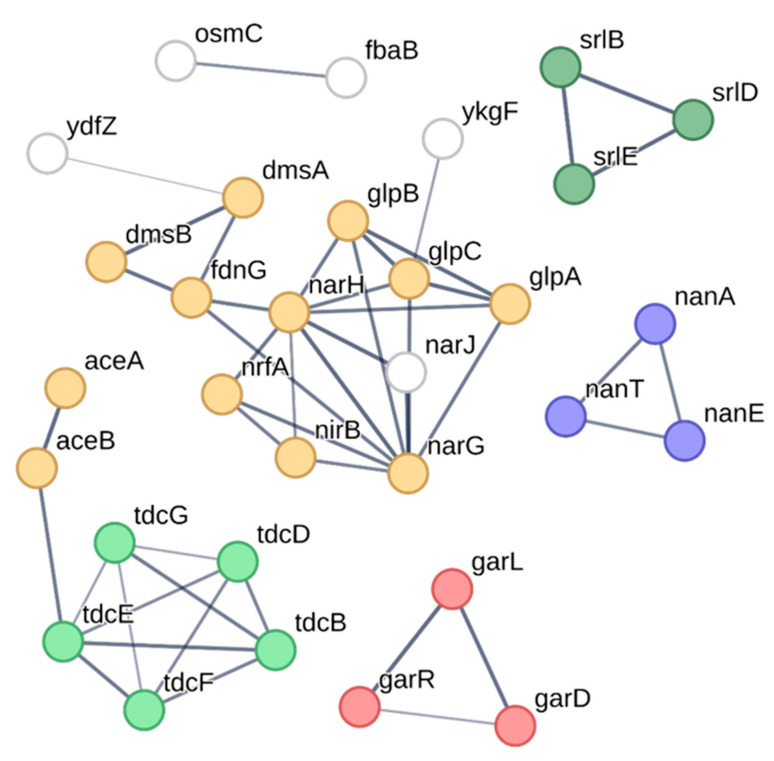
STRING image showing the interaction among the DEP proteins. The colored circles indicate the proteins that belong to the same biological process. Anaerobic respiration (primrose yellow); sorbitol transport (green); L-threonine catabolic process to propionate (pale green); mannosamin metabolic process and sialic acid transport (purple); galactarate metabolic process (red).

**Table 1 ijms-24-14009-t001:** MDR plasmids used in this study.

Name	pCTXM123_C0996	pCTXM64_C0967	pHK01	pNDM-HK	pNDM-HN380	pJIE143
Inc Group	I1	I2	FII	M2	X3	X4
Size	108 Kb	62 Kb	70 Kb	89 Kb	54 Kb	34 Kb
Source	*Enterobacteriaceae*	*Enterobacteriaceae*	*Enterobacteriaceae*	*Enterobacteriaceae*	*Enterobacteriaceae*	*Enterobacteriaceae*
Resistance Gene	*bla* _CTX-M-123_	*bla* _CTX-M-64_	*bla* _CTX-M-14_	*bla* _TEM-1_ *bla* _NDM-1_ *bla* _DHA-1_ *sul1* *qacdelta1*	*bla* _NDM-1_ *bla* _SHV-12_ *ble* _MBL_	*bla* _CTX-M-15_
CTX Group	CTX-M-1/9 group hybrids	CTX-M-1/9 group hybrids	CTX-M-9 group	-	-	CTX-M-1 group

## Data Availability

Not applicable.

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
