# Peer review of "Comparative Analysis of Transcriptome and Proteome Revealed the Common Metabolic Pathways Induced by Prevalent ESBL Plasmids in *Escherichia coli"

_ijms, 2023, doi:10.3390/ijms241814009_

Round 1
Reviewer 1 Report
The manuscript titled "Comparative Analysis of Transcriptome and Proteome Reveals Common Metabolic Pathways Induced by Prevalent ESBL Plasmids in E. coli" provides an in-depth investigation into the influence of multidrug resistance (MDR) plasmids on the gene expression and metabolic pathways of E. coli J53 transconjugants. The authors emphasize the significance of plasmids as critical agents in the dissemination of drug-resistant genes, while highlighting the need for a comprehensive understanding of their complete functional roles beyond drug resistance.
Here are the critiques:
- The manuscript provides valuable insights into the impact of MDR plasmids on bacterial gene expression, metabolic pathways, and adaptation. These findings contribute to a broader understanding of the complex interplay between plasmids and bacterial hosts, shedding light on how plasmids influence bacterial survival and growth. However, the manuscript could explicitly emphasize the novelty of these findings within the context of existing literature and discuss how they advance our knowledge of MDR plasmid effects beyond previous studies. Furthermore, functional annotation of genes will enhance the quality of the paper.
- While the manuscript identifies differentially expressed genes influenced by MDR plasmids, the functional interpretation of these genes in bacterial fitness and antibiotic resistance context is somewhat limited. Providing more detailed insights into how these gene expression changes directly relate to the observed phenotypic effects would enhance the depth of the discussion.
- While the manuscript mentions potential therapeutic targets, it does not extensively discuss the clinical relevance of the findings. Addressing how the identified pathways and genes may impact clinical outcomes and treatment strategies would enhance the manuscript's practical implications.
Reviewer 2 Report
The background for this paper is plasmids that code for multidrug resistance have become a problem, and there are a lot of them. From the introduction: “little is known about the prevalent MDR plasmids and their interaction to the bacterial genome, in particular the survival advantages confer to the bacteria in addition to providing drug resistance…[D]iscovery of the common MDR plasmid-induced pathways in the bacterial host which were crucial for the survival of drug-resistant bacteria…as druggable targets…and adaptation of bacteria in different environments.” The results show growth rates with the conjugal plasmids, transcriptomics, computational metabolic pathway analysis, proteomics, and carbon utilization. The conclusions were that the MDR plasmids increased TCA cycle enzymes, electron transport components, carbohydrate utilization pathways; changes in the transcriptome and proteome were not congruent (leading to the proposal of major translational control); the MDR plasmids reprogram growth, and rpoS expression increases in log phase.
The results have alternate explanations that are not consistent with the stated conclusions, in addition to problems with the strength of the conclusions.
Major concerns:
1. The point of this paper is that the characteristics of the MDR plasmids are specific to MDR plasmids. It is possible that these characteristics are the results of almost any (not necessarily all) plasmids. With this in mind, plasmids that are not MDR should have been used as controls. As it is, the only control is the parental strain without a plasmid. No statement about the properties of MDR plasmids can be made without appropriate plasmid controls.
2. All experiments were done with LB medium. There is nothing inherently wrong with the medium for which growth is based on amino acids as the energy source. The TCA cycle and several genes of anaerobic metabolism are induced with the plasmids. If the burden of plasmid carriage makes the medium more anaerobic (probably) and more alkaline (probably) then the differences in metabolism are not specific to MDR plasmids. In addition to differences in oxygen availability (and in this context the rpm of shaking is not given), it seems likely that elevated levels of cyclic AMP can account for most of these differences in metabolism. High cAMP can account for higher TCA cycle activity and increased expression of carbohydrate catabolic pathways.
3. The differences in growth rates with and without plasmids (aside from requiring control plasmids) (Figure S1) is really important and requires statistical validation which in this case means more replication.
4. The charts with gene names are unreadable, even using computer-aided magnification. This problem made review verging on the impossible.
5. Numerous examples of misstating of basic facts or small problems occurs. A partial list follows: line 160, describes a one-fold change, when a two-fold change was probably intended; references to Figure 3 ask the reader to go to the wrong subsection (e.g., line 166); the glpABC and glpD operons are merged into one operon, which is not correct; lines 133 suggests that glpD is essential except that it is essential only when glycerol is the carbon source, not generally essential. The last comment suggested to the reviewer that more references needed to be checked to determine if the reference states what the author intended.
6. The introduction can include a brief summary of previous studies that analyzed the effects of plasmids (non-MDR plasmids) on the topic of this paper.
The problems with English usage are not major and can be fixed at the journal, if accepted.
Round 2
Reviewer 1 Report
The authors have revised all of the suggestions.
